# Genome-Wide Identification of Freezing-Responsive Genes in a Rapeseed Line NTS57 Tolerant to Low-Temperature

**DOI:** 10.3390/ijms252312491

**Published:** 2024-11-21

**Authors:** Guodong Zhao, Jiaping Wei, Junmei Cui, Shichang Li, Guoqiang Zheng, Zigang Liu

**Affiliations:** State Key Laboratory of Aridland Crop Science, Gansu Agricultural University, Lanzhou 730070, China; 1073324120394@st.gsau.edu.cn (G.Z.); cuijm@gsau.edu.cn (J.C.); 1073324120375@st.gsau.edu.cn (S.L.); zhenggq@st.gsau.edu.cn (G.Z.)

**Keywords:** winter rapeseed, freezing stress, transcriptome, differentially expressed genes, heat stress transcription factor, heat shock protein

## Abstract

Winter rapeseed is a high-oil crop that exhibits significant sensitivity to low temperatures, leading to a substantial reduction in production. Hence, it is of great significance to elucidate the genomic genetic mechanism of strong freezing-resistant winter rapeseed to improve their freezing-resistant traits. In this study, global transcriptome expression profiles of the freezing-resistant cultivar NTS57 (NS) under freezing stress were obtained for the years 2015, 2016, and 2017 by RNA sequencing (RNA-seq). Most differentially expressed genes (DEGs) were involved in the plant hormone signal transduction, alpha-linolenic acid metabolism, protein processing, glutathione metabolism, and plant-pathogen interaction pathways. Antioxidant enzyme activities and lipid peroxidation levels were significantly positively and negatively correlated with overwintering rate (OWR), respectively. After freezing treatment, the formation of freezing resistance of NS was attributed to the increase in antioxidant enzyme activities and content of osmotic regulation substances, as well as the decrease in lipid peroxidation level. Furthermore, quantitative reverse transcription polymerase chain reaction (qRT-PCR) and phenotypic verification indicated that heat stress transcription factor A2 (*HSFA2*) and 17.6 kDa class II heat shock protein (*HSP17.6*) participated in the response to freezing stress. This study will further refine the regulatory network of plants against freezing stress and help to screen candidate genes for improving plant freezing resistance.

## 1. Introduction

Freezing stress (<0 °C) is a common and destructive environmental stress in Northwest China, which not only restricts the geographical location of plants but also adversely affects plant growth and development as well as agricultural productivity [1,2]. In addition, freezing stress can induce a cascade of physiological and biochemical alterations, such as membrane lipid peroxidation, reactive oxygen species (ROS) increase, and cellular structural destruction [3]. In response to freezing stress, plants have evolved complex adaptive mechanisms that improve their freezing tolerance when exposed to freezing stress [4,5].

Winter rapeseed (*Brassica napus* L.) is an important oilseed crop that is widely cultivated in Northwest China, but it is susceptible to low temperatures in winter. Freezing temperature actually prevents the winter rapeseed from overwintering and propagating, resulting in a huge decline in production. Over the past two decades, much progress has been made in identifying the crucial components that participate in cold stress tolerance and uncovering their regulatory mechanisms. For example, a cold sensor CHILLING-TOLERANCE DIVERGENCE 1 (COLD1) was reported to regulate a cold-sensing calcium channel in rice leading to the activation of the cold-regulated genes [6]; the C-repeat binding factors (CBFs), as the dehydration responsive element binding factors, are critical for the cold acclimation in higher plants [7,8]; Inducer of CBF Expression 1 (ICE1) is a key regulator of cold, and OPEN STOMATA 1 (OST1) kinase modulates freezing tolerance by enhancing ICE1 stability in *Arabidopsis* [9]. Despite considerable interest over several decades, most researchers working on cold tolerance focused only on chilling temperatures (>0 °C), while the mechanism of winter crops in freezing tolerance remained very poorly understood. *Brassica napus* NTS57 is a freezing-tolerant cultivar that is derived from a distant cross between *Brassica napus* and *Brassica rapa*. It has strong cold resistance and can survive in the field at −26 °C in northwest China [3]. Hence, the knowledge of the freezing resistance genes and the molecular regulation mechanism of winter rapeseed NTS57 will provide valuable information and genetic resources for improving freezing stress tolerance in crops.

In this study, a correlation analysis between the OWR and the physiological changes was evaluated in the freezing-resistant cultivar NTS57. The transcriptome expression profiles of the NTS57 during overwintering were obtained. Combining the physiological data and transcriptome data, we identified a suite of DEGs related to freezing stress, and subsequently investigated the metabolic pathways enriched by these DEGs. Furthermore, hundreds of transcription factors (*TFs*) were found to be differentially expressed, many of which were studied in depth to contribute to cold stress in plants. Finally, the functions of *HSFA2* and *HSP17.6* were characterized under cold stress. These findings are intended to provide essential insights into the underlying mechanisms of cold tolerance in winter rapeseed under freezing stress.

## 2. Results

### 2.1. Effects of OWR on Plant Morphological, Physiological, and Biochemical Characteristics in Winter Rapeseed NTS57

The freezing tolerance of the winter rapeseed NTS57 was evaluated based on its OWR, which exhibited a significant year-on-year increase, ultimately reaching 72.3%, 85.3%, and 89.5% in the 2015-NTS57 (15NS), 2016-NTS57 (16NS), and 2017-NTS57 (17NS) plants, respectively (Figure 1A). Plant morphology observation showed that the 17NS plants were the strongest after the freezing treatment followed by the 16NS plants, whereas the 15NS plants were the weakest (Figure 1B).

The antioxidant enzymatic (peroxidase, POD; superoxide dismutase, SOD; catalase, CAT) activities significantly increased in the NTS57 leaves year by year (Figure 2A–C) and they showed a significant positive correlation with OWR (Table 1). The soluble sugar and free proline content of the osmotic substances in the NTS57 leaves has significantly increased over the years, while the soluble protein content has significantly decreased (Figure 2D–F). In addition, the level of lipid peroxidation in the NTS57 leaves was evaluated by determining malondialdehyde (MDA) and relative electrolytic leakage (REL). Over the years, the MDA content and REL declined significantly in the NTS57 leaves (Figure 2G,H) and there was a significant negative correlation with the OWR (Table 1). These results highlight that the antioxidant enzyme activities, osmotic regulation substances, and lipid peroxidation level may be involved in the development of freezing tolerance.

### 2.2. Quantitative Transcriptomic Analysis of Winter Rapeseed Under Freezing Stress

The PCA analysis demonstrated that the three biological replicates of 15NSt0, 16NSt0, 17NSt0, 15NSt2, 16NSt2, and 17NSt2 exhibited strong conformity (Figure 3A and Appendix A), and their correlation coefficient exhibited a value exceeding 0.9 (Figure 3B). This indicates a substantial level of reliability in the transcriptome analysis.

To identify the DEGs that were induced by the freezing treatment, the changes at the transcription level in the leaves of NS were analyzed by calculating the relative level ratio of treatment to control. After the freezing treatment, a total of 4421, 6611, and 6253 DEGs were identified to be up-regulated in the leaves of 15NS, 16NS, and 17NS, respectively, whereas 9429, 7107, and 10,230 DEGs were down-regulated in 15NS, 16NS, and 17NS, compared to controls (Figure 4A and Appendix A). The number of DEGs was significantly different among 15NS, 16NS, and 17NS, implying that the dynamics of the gene expression regulation in the three consecutive years were different. Furthermore, 3138, 4295, and 4761 DEGs were specifically identified only in 15NS, 16NS, and 17NS (Figure 4B and Appendix A). Additionally, 5669 shared DEGs were present in 15NS, 16NS, and 17NS. Among the commonly identified DEGs, 2662 were found to be up-regulated, while 2979 exhibited down-regulation (Figure 4B and Appendix A), which may be core molecular components in NS in response to freezing stress. These DEGs were considered as the candidate genes underlying freezing resistance and used for further analysis. A notable positive correlation was identified between the fold changes of the common DEGs, for example, a coefficient of 0.897, 0.920, and 0.909 was noted in 15NS_16NS, 15NS_17NS, and 16NS_17NS, respectively (Figure 4C–E).

### 2.3. Gene Ontology (GO) Annotation and Kyoto Encyclopedia of Genes and Genomes (KEGG) Pathway Analysis of DEGs Under Freezing Stress

All the DEGs identified in 15NS, 16NS, and 17NS were subjected to a GO classification and KEGG pathway analysis. In total, hundreds of metabolic pathways were modified in the leaves of 15NS, 16NS, and 17NS when subjected to freezing stress (Appendix A). Among them, the cellular process, cell, and binding were the largest groups in the biological process, cellular component, and molecular function categories, respectively (Appendix A). The DEGs in 15NS were significantly (*q*-value < 0.001) enriched in plant hormone signal transduction (ko04075), photosynthesis (ko00195/ko00196), alpha-linolenic acid metabolism (ko00592), biosynthesis of secondary metabolites (ko01110), and glutathione metabolism (ko00480) (Figure 5A). The DEGs in 16NS were significantly (*q*-value < 0.001) enriched in plant hormone signal transduction (ko04075), alpha-linolenic acid metabolism (ko00592), linoleic acid metabolism (ko00591), biosynthesis of secondary metabolites (ko01110), and plant–pathogen interaction (ko04626) (Figure 5B). The DEGs in 17NS were significantly (*q*-value < 0.001) enriched in plant hormone signal transduction (ko04075), alpha-linolenic acid metabolism (ko00592), biosynthesis of secondary metabolites (ko01110), photosynthesis (ko00195/ko00196), and plant–pathogen interaction (ko04626) (Figure 5C). In addition, we found that the protein processing pathway in endoplasmic reticulum (ko04141) enriched a substantial number of DEGs (Appendix A). These metabolic pathways are regarded as the potential pathways implicated in the freezing tolerance of winter rapeseed.

### 2.4. Differentially Expressed TFs Under Freezing Stress

Totally, 530 differentially expressed *TFs* were screened from the candidate genes within the leaf transcriptome of NS exposed to freezing stress (Appendix A). Among these, the majority of the up-regulated differentially expressed *TFs* were classified as ethylene response factor (AP2/ERF), NAC domain-containing protein (NAC), WRKY transcription factor (WRKY), TIFY protein (TIFY), *HSF*, MADS-box protein (MADS), and TAZ domain-containing protein (TAZ). The predominant portion of the down-regulated differentially expressed *TFs* were categorized as belonging to the bHLH (bHLH), trihelix, C2C2-GATA, mMTEF, and squamosa promoter-binding-like (SBP) transcription factor families (Figure 6). It is worth noting that all the *HSFs* were significantly up-regulated after freezing stress. These findings suggest that TFs, especially HSFs, were closely related to freezing stress.

### 2.5. RNA-Seq Validation Through qRT-PCR

The qRT-PCR analysis was employed to assess the reliability of the transcriptomic data. Eight DEGs responsible for the freezing response in NTS57 were selected as targets. The expression pattern of all DEGs exhibited complete consistency between mRNA and RNA-seq levels, and a significant positive correlation with a coefficient of 0.386 was noted (Figure 7). It underscored the acceptability of RNA-seq in the present study.

### 2.6. The BnaHSFA2 and BnaHSP17.6 Transport into Yeast Cells Showed Improved Cold Tolerance

To further identify whether the candidate genes are able to respond to freezing stress, the *BnaHSFA2* and *BnaHSP17.6* were transformed into yeast cells; all transformations were grown in the SD-Ura solid medium. The yeast phenotypic analysis showed that the growth of cells carrying pYES2 (control) was more severely affected compared to the pYES2-*BnaHSFA2* at 4, 0, −4, −10, and −20 °C, respectively, and the cells carrying pYES2 basically did not grow after 72 h and 96 h of treatment at −20 °C (Figure 8). The cells carrying pYES2 were more sensitive to low temperatures (4, 0, −4, −10, and −20 °C) than the pYES2-*BnaHSP17.6*, which grow well after 72 h and 96 h of treatment at −20 °C (Figure 9). These results suggest that *BnaHSFA2* and *BnaHSP17.6* can improve cold tolerance in yeast cells.

## 3. Discussion

### 3.1. Physiological Changes During Overwintering in Winter Rapeseed NTS57

Cold stress can cause a series of physiological changes in plants [10]. It is well known that POD and SOD are two crucial antioxidant enzymes responsible for ROS scavenging to maintain normal cellular redox homeostasis [11]. Previous studies proved that the cold-acclimated rapeseed leaves increased the activities of antioxidant enzymes [12]. Amini et al. [13] suggested that the relative electrolyte leakage index decreased in cold-tolerant chickpeas under cold stress. Hussain et al. [14] reported that the cold treatment induced a decline in MDA content in the leaves of cold-tolerant soybean. After cold treatment, the transgenic *Arabidopsis* was demonstrated to contain higher free proline, lower MDA, and relative electrolyte leakage than the WT plants [15]. In this study, for plants with a high OWR, we observed an elevated activity of antioxidant enzymes accompanied by a decreased magnitude of lipid peroxidation. The SOD activity of 17NS was found to be 3.7 and 2.8 times greater than that of 15NS and 16NS, respectively, while the MDA content of 15NS and 16NS was 4.2 and 3.6 times higher than that of 17NS, respectively (Table 2). Soluble sugars and free proline function as the osmotic regulation substances to protect plants from damage from cold stress [16]. Recent scientific advancement reported that the combined drought and cold stress induced the accumulation of proline in maize metabolomics [17] and also caused the accumulation of soluble sugars in potato tubers [18]. Our results show that cold stress prompted the accumulation of proline and soluble sugar in the winter rapeseed NS plants, and their cold resistance was gradually enhanced (Table 2). In brief, antioxidant enzymes and osmotic regulation substances might be involved in ROS scavenging and cellular oxidative homeostasis, which might contribute to enhanced stress tolerance.

### 3.2. Transcription Level Changes During Overwintering in Winter Rapeseed NTS57

A large spectrum of transcription factors has been identified to be involved in cold stress signals and promoting the expression of the downstream cold-responsive genes, thus enhancing freezing tolerance [5,19,20]. The dehydration-response element-binding protein CBFs are principal components responsible for cold-responsive gene expression [7,21,22]. The *CBF* expression is induced immediately after the cold treatment and is both positively and negatively regulated by the various transcription factors, such as calmodulin-binding transcription activator (CAMTA1-5), brassinazole-resistant1 (BZR1), and heat shock transcription factor A1 (HSFA1), which are all positive regulators of the CBF expression and freezing tolerance [23,24,25]. On the contrary, MYB transcription factor 15 (MYB15), ethylene insensitive 3 (EIN3) and phytochrome interacting factor 3/4/7 (PIF3/4/7) inhibit the expression of *CBF* and negatively regulate freezing tolerance [26,27,28]. In this study, the majority of *AP2/ERF*, *MADS*, *HSF*, *NAC*, *TIFY*, *TAZ*, and *WRKY* transcription factors exhibited up-regulation in NTS57; conversely, *bHLH*, *Trihelix*, *C2C2-GATA*, *mTERF*, and *SBP* demonstrated down-regulation (Figure 6). This implies that the regulation of gene expression is intricate, and these *TFs* are more likely to be activated in order to modulate the expression of their corresponding target genes in response to freezing stress.

There is increasing evidence that plant hormones interact with cold signaling to adapt well to cold stress [29,30,31]. In this study, a substantial number of DEGs were significantly enriched in the plant hormone signaling pathway (Appendix A), of which those involved in jasmonate (JA) signaling were up-regulated. They encoded TIFY protein and jasmonic acid amido synthetase, which implies that the JA synthesis was increased. It is in agreement with the findings reported by [30], who indicated that a high level of JA could improve freezing tolerance. The DEGs engaged in the ABA signaling, specifically those encoding protein phosphatases 2C and serine/threonine-protein kinases, were found to be up-regulated, whereas the genes encoding the majority of abscisic acid receptors exhibited down-regulation; two DEGs (one up-regulation and one down-regulation) encoding EIN3, were found to be associated with ethylene signaling; in the auxin signaling pathway, most of the DEGs encoding auxin-responsive/transporter protein and PIF4 were down-regulated, while five out of six DEGs encoding indole-3-acetic acid-amido synthetase (GH3) were up-regulated; this asserted that there may be two distinct metabolic pathways involving ABA, ethylene, and auxin that contribute to freezing tolerance, with one of these pathways being up-regulated. Moreover, a down-regulated *BZR1*, recognized as a plant-specific positive regulator involved in brassinosteroid signaling, was accompanied by the low expression of BRI1-associated receptor kinase1. This implies that the BRI1-associated receptor kinase1 might cooperate with BRI1 and positively modulate *BZR1*, which echoes the previous findings [24,32]. In addition, three DEGs encoding gibberellin (GA) receptor GID1 and DELLA protein were up-regulated. We hypothesized that GID1 might interact with DELLA to stimulate the expression of freezing-responsive genes in the GA signaling pathway, as in the previous report [33]. The response regulators (RRs) are typically categorized into two distinct types, namely type A and type B. The type-A RRs serve as negative modulators of cytokinin (CK) signaling, whereas the type-B RRs act as positive modulators of CK signaling [34]. Correspondingly, the five RRs identified in this investigation were encoded by the *ARR8*, *ARR9*, and *ARR14* genes, and their expression was found to be down-regulated. Yet another novel finding is the down-regulation of histidine-containing phosphotransfer protein (HPP), a signal sensor that facilitates the transfer of phosphate from a receptor kinase to a RR in the nucleus [35]. This meant that ARR8, ARR9, and ARR14 are type-A RRs; they may participate in the negative feedback regulation of the CK signaling pathway in conjunction with HPP following freezing treatment. Collectively, these findings suggest that the plant hormones used to play critical roles in the response of winter rapeseed to freezing stress.

Linoleic acid and α-linolenic acid are the two vital unsaturated fatty acids that are integral in the maintenance of cell membrane integrity under cold stress [36]. Alpha-linolenic acid is known as a biosynthetic precursor of JA [37]. In this study, we found that the several key enzymes of JA synthesis, lipoxygenase (LOX), allene oxide synthase (AOS), 12-oxophytodienoate reductase (OPDR), and allene oxide cyclase (AOC), were up-regulated (Appendix A and Appendix A). It appeared that the jasmonic acid synthesis was enhanced in NTS57 under freezing stress. This aligns with the findings from hormonal signaling studies. On the same line, previous researchers have reported that the unsaturated fatty acid content rose in chickpeas under cold stress [38].

Heat shock proteins (HSPs) are recognized as the primary targets of HSFs and play a crucial role in mediating cold-stress responses in plants [39,40]. A suite of studies have shown that HSFs can be triggered by reactive oxygen species signals, followed by the mitigation of oxidative damage by enhancing the antioxidant enzyme activities [41,42]. In this study, all the DEGs encoding HSP70/90/17.6/18.1/22/23.5/25.3, protein transport protein, cell division control protein, chaperone protein, and nucleotide exchange factor enriched in the protein processing pathway were up-regulated (Appendix A and Appendix A). Additionally, ten up-regulated DEGs were found to encode the nucleus-localized RNA polymerase II transcription mediator, which is a member of the *HSP70* family and could activate the expression of cold-regulated genes under cold stress [43]. All the *HSFs* displayed consistent expression patterns with the *HSPs* (Figure 6 and Appendix A). Furthermore, we also found some down-regulated DEGs were significantly enriched in this pathway (Appendix A and Appendix A), suggesting the potential involvement of post-transcriptional modifications in these DEGs. Glutathione S-transferases (GSTs) are an important target for research on plant stress tolerance mechanisms [44]. Glutathione peroxidases (GPXs) and ascorbate peroxidases (APXs) are central components of ROS scavenging mechanisms and participate in the maintenance of membrane integrity [45,46]. In the present study, we found a large number of up-regulated DEGs enriched in the glutathione metabolism pathway, which encode GST, GPX, and APX (Appendix A and Appendix A), suggesting that enhanced ROS scavenging may maintain the stability of cell morphology in the leaves of winter rapeseed. The above results are integrated with elevated levels of antioxidative enzymatic activities (Figure 2) from the physiological measurement, conveying that the ROS signals were unleashed immediately after being subjected to freezing stress. Subsequently, HSFs interact with HSPs to regulate the expression of the downstream freezing-responsive genes.

Recent reports have demonstrated that cold stress triggers the expression of plasma membrane-located receptors and calcium signaling components, such as mitogen-activated protein kinase (MAPK), Ca^2+^-dependent protein kinase (CDPK), calcium/calmodulin-regulated receptor-like kinase (*CRLK*), receptor-like kinase (*RLK*), calcium-binding protein CML (CML), and calmodulin (CaM) [47,48]. Likewise, several *CDPKs*, *MAPKs*, *CMLs*, *CaMs*, and PTI1-like tyrosine-protein kinases (*PTKs*) were found to be up-regulated and linked to the pathways involved in plant pathogen interaction. Intriguingly, all the *WRKYs* and *HSPs* enriched in this pathway were also up-regulated (Appendix A and Appendix A). These findings imply that the Ca^2+^ sensors in rapeseed might detect the cold stimulus subsequent to freezing treatment and initiate the Ca^2+^ signaling cascade. Consequently, the accumulated Ca^2+^ influx activates some plasma membrane protein kinases to interact with WRKYs and promote the expression of *HSPs* to adapt to the freezing environment. Similarly, a previous study has also revealed that high levels of Ca^2+^ sensors and HSPs were induced in winter rapeseed under freezing stress [3]. Additionally, certain DEGs involved in plant–pathogen interaction encoding disease resistance protein (DRP) and pathogenesis-related protein (PRP) were down-regulated (Appendix A and Appendix A). At the same time, two down-regulated *PRPs* were found to be enriched in the salicylic acid (SA) signaling pathway. Therefore, we speculated that SA is more likely to interact with plant pathogens to negatively regulate freezing tolerance.

### 3.3. BnaHSFA2/BnaHSP17.6 Enhanced the Cold Tolerance in Winter Rapeseed NTS57

*HSF TFs* are a superfamily with 64 members in *Brassica napus* [49]. Many studies have shown that there are *HSF* transcriptional cascades in plant stress responses. The transcriptional regulatory cascades *HSFA1-HSFA2-HSPs* and *HSF3-EDS1-PR* are involved in the resistance to thermal stress [50] and resistance to disease [51], respectively. Yeast has been effectively employed as a model for exploring the function of plant genes [52]. Hence, we used a yeast system to examine the relationship between the *BnaHSFA2*/*BnaHSP17.6* and cold response. The pYES2-*BnaHSFA2-* and pYES2-*BnaHSP17.6*-transformed BY4717 strains grew better than their corresponding pYES2-transformed control under cold stress (Figure 8 and Figure 9), implying that *BnaHSFA2* and *BnaHSP17.6* might cascade to participate in the cold stress response in yeast. These results are consistent with those of qRT-PCR (Figure 7).

## 4. Materials and Methods

### 4.1. Planting and Freezing Stress Treatments

The plant material, the winter rapeseed cultivar NTS57 known for its freezing resistance and extensively cultivated in Northwest China, was furnished by Gansu Agricultural University [3]. The NTS57, cultivated in the years 2015, 2016, and 2017, was planted in plastic pots filled with a 3:1 ratio of nutrient soil to vermiculite and labeled, in short, as 15NS, 16NS, and 17NS, respectively. When the potted plants had four leaves, they were delicately segregated into two groups. The treatment group (Treatment, t2) was placed in a chamber at −4 °C with 60% humidity and a 12 h/12 h (day/night) photoperiod for 24 h, while the control group (Control, t0) was kept in normal conditions of 22 °C, 60% humidity, and a 12 h/12 h (day/night) photoperiod for 24 h. The second leaf was collected from both the control and treated plants, with each sample comprising three biological replicates. The harvested samples were promptly frozen in liquid nitrogen and subsequently stored in a −80 °C refrigerator for total RNA extraction.

### 4.2. Evaluation of OWR and Physiological Characteristics

In the present study, the seeds of NTS57 were sown in holes in August of 2015, 2016, and 2017, respectively, and the OWR was numbered after the rapeseed had turned green in the following year. The OWR is defined as the ratio of the actual number of seedlings that successfully emerge to the total number of seedlings that were originally planted. A total of 0.2 g of rapeseed leaves from the samples labeled as 15NS, 16NS, and 17NS were utilized to assess various physiological and biochemical indices, including POD, SOD, CAT, free proline, soluble sugar, soluble protein, MDA, and REL. The POD, SOD, and CAT activities were determined as described by [53]. The proline content was measured based on the previous study [3]. The contents of soluble sugar and soluble protein were determined as described by [54]. The degree of lipid peroxidation was determined by quantifying the content of MDA as described by Wei et al. [55]. The REL was determined as described by Huang et al. [56]. All experiments were repeated three times.

### 4.3. RNA Extraction and RNA Sequencing

Total RNA was extracted from six samples, each comprising three biological replicates, which were ground in liquid nitrogen and processed using the TRIzol Reagent (Tiangen Biotech, Beijing, China) in accordance with the manufacturer’s guidelines. The construction of the library and subsequent sequencing were conducted by Gene Denovo Biotechnology Co., based in Guangzhou, China, utilizing the Illumina HiSeq™ 2500 platform. Following the Illumina sequencing, the raw sequences from three biological replicates of each sample were subjected to filtering processes to yield clean reads for subsequent analyses.

### 4.4. Filtering of Reads and Analysis of Gene Expression

High-quality clean data were acquired following the removal of reads containing adapters or poly-N sequences and low-quality reads. The high-quality paired-end reads from each sample were aligned to the rapeseed reference genome using TopHat v2.0.3.12 [57]. The gene expression levels were quantified and standardized as fragments per kilobase of transcript per million mapped reads (FPKM), which can be directly employed for the identification of DEGs in pairwise comparisons [58].

Genes exhibiting a false discovery rate (FDR) of <0.001 and an absolute value of |log2FoldChange| ≥ 2 by the edgeR package (http://www.r-project.org/, accessed on 22 October 2024) were defined as the DEGs. The raw data of the sequenced transcriptome have been deposited in the SRA at NCBI under accession number PRJNA685002.

### 4.5. qRT-PCR Analysis

The expression patterns of the twelve candidate genes responsive to freezing stress were evaluated using qRT-PCR. The qRT-PCR was executed in accordance with the previous protocol [3]. All the primers are presented in Appendix A. The relative quantification (2^−ΔΔCt^) of gene expression was evaluated through a comparative cycle threshold approach, and each sample was replicated thrice.

### 4.6. Construction of Yeast Transformation Vector and Phenotypic Analysis

The nucleotide sequences of *BnaHSFA2* and *BnaHSP17.6* were obtained according to their accession numbers in Genebank, XM_013879521.3 and XM_013860746.3 (Brassica napus heat stress transcription factor A-2-like, *BnaHSFA2* and Brassica napus 17.6 kDa class II heat shock protein, *BnaHSP17.6*). The open reading frames of *BnaHSFA2* and *BnaHSP17.6* were inserted into *EcoR* I and *BamH* I sites of pYES2 (Ruiyuan, Nanjing, China) for constructing integrated vectors pYES2-*BnaHSFA2* and pYES2-*BnaHSP17.6* (Appendix A). The recombinant vector plasmids and pYES2 (control) were transformed into BY4717, and positive transformants were used for the cold tolerance analysis on synthetic medium without uracil (SD/-Ura, SD-U) after cold stress treatment (4, 0, −4, −10, and −20 °C) for 24, 48, 72, and 96 h, respectively.

### 4.7. Bioinformation Analysis

The inter-replicate correlation coefficient was computed to assess the repeatability of the experimental results among samples. The principal component analysis (PCA), Pearson’s correlation coefficient heat map, and cluster expression heat map were carried out to disclose the structure/relationship of samples employing an R package gmodel (http://www.r-project.org/, accessed on 22 October 2024). The genes were annotated with reference to the GO database and the KEGG database [59,60]. Enrichment analysis was executed in accordance with the GO and KEGG databases. The t-test was employed to perform an analysis of the significant variations in physiological data with a confidence interval of 95% or 99%.

## 5. Conclusions

In summary, our research successfully established a comprehensive global transcriptome expression profile of winter rapeseed cultivar NTS57 in response to freezing stress and identified over 5000 DEGs, which were predominantly concentrated in the regulation of plant hormone signal transduction, alpha-linolenic acid metabolism, protein processing, and plant–pathogen interaction pathways. Moreover, the *BnaHSFA2* and *BnaHSP17.6* were involved in the response to freezing stress.

## Figures and Tables

**Figure 1 ijms-25-12491-f001:**
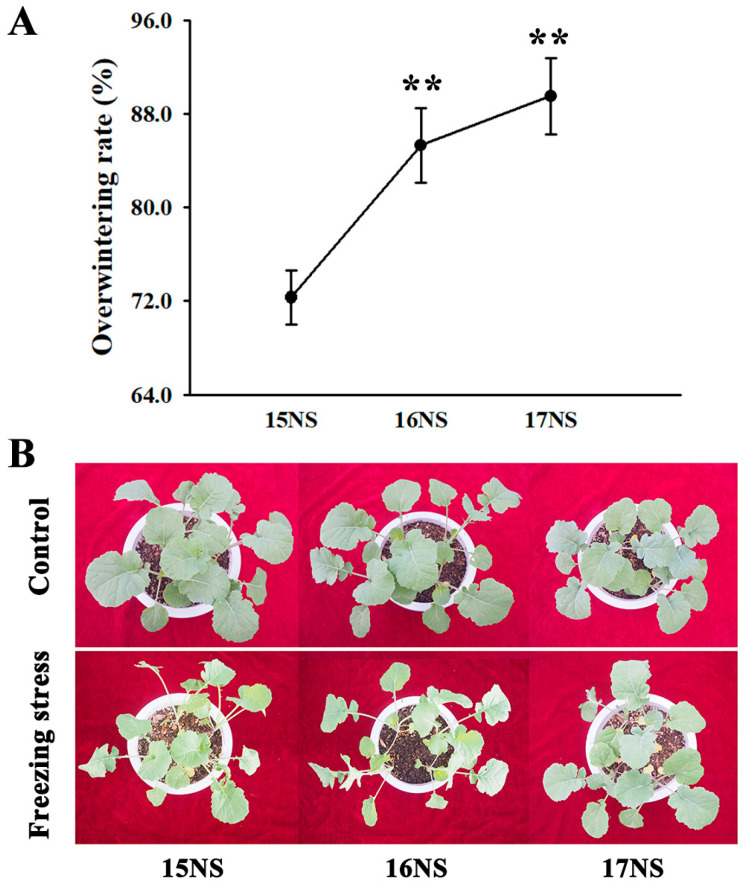
The OWR and morphological adaptations of winter rapeseed under freezing stress. (**A**) OWR of the NS plants cultivated in 2015, 2016, and 2017. The values presented are expressed as means ± standard deviation derived from three biological replicates (**, *p* < 0.01); (**B**) The 5-week-old NS plants subjected to freezing treatment and cultivated in the years 2015, 2016, and 2017.

**Figure 2 ijms-25-12491-f002:**
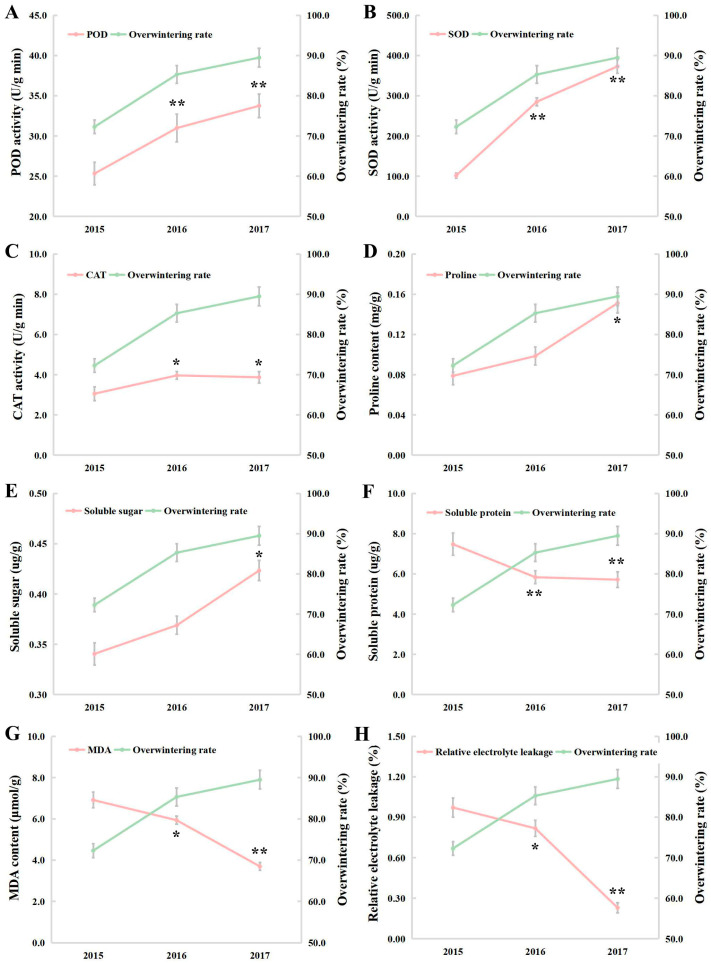
Effects of OWR on physiological and biochemical changes in the winter rapeseed seedlings cultivated in 2015, 2016, and 2017. (**A**) POD; (**B**) SOD; (**C**) CAT; (**D**) proline; (**E**) soluble sugar; (**F**) soluble protein; (**G**) MDA; (**H**) REL. The values are means ± standard deviation from three biological replicates (**, *p* < 0.01; *, *p* < 0.05).

**Figure 3 ijms-25-12491-f003:**
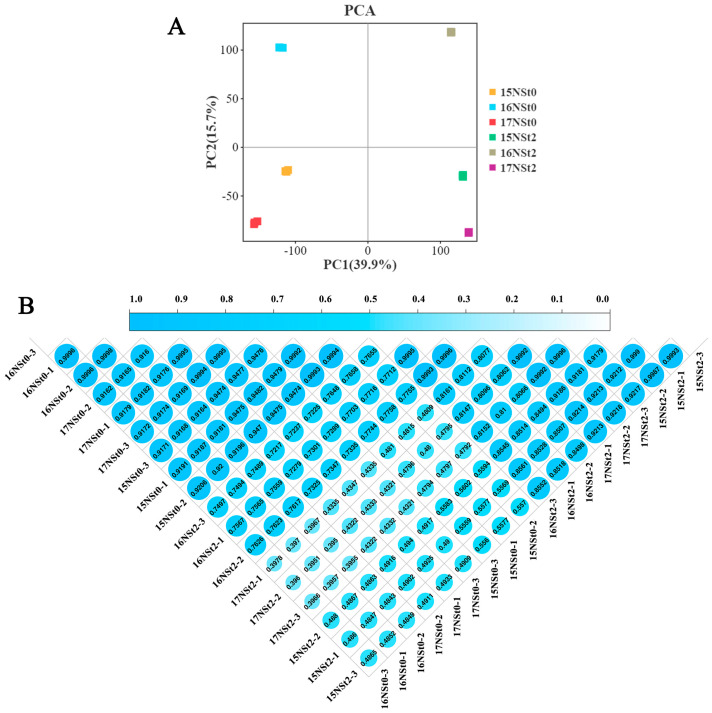
Overview of transcriptome in leaves of winter rapeseed NTS57. (**A**) A PCA of the detected genes by the RNA−seq. (**B**) Pearson’s correlation coefficient heat map among the three biological replicates of 15NSt0, 16NSt0, 17NSt0, 15NSt2, 16NSt2, and 17NSt2 samples.

**Figure 4 ijms-25-12491-f004:**
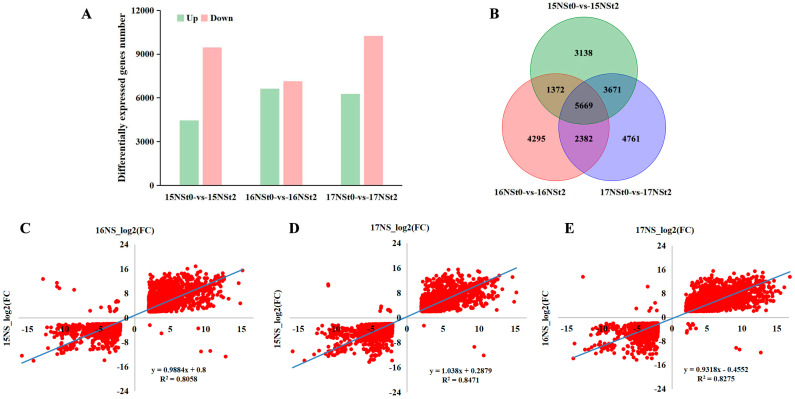
Comparison of transcript abundance in leaves of winter rapeseed NTS57. (**A**) The number of DEGs identified in 15NS, 16NS, and 17NS under freezing stress. (**B**) A Venn diagram of identified DEGs in 15NS, 16NS, and 17NS under freezing stress. The correlation between the expression patterns of the shared DEGs in (**C**) 15NS−16NS, (**D**) 15NS−17NS, and (**E**) 16NS−17NS, respectively.

**Figure 5 ijms-25-12491-f005:**
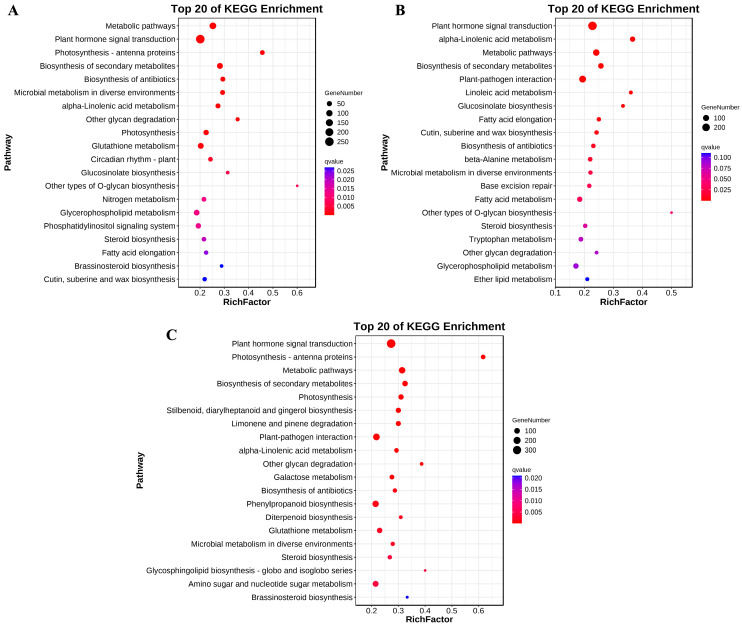
The top 20 KEGG pathways enriched by DEGs in 15NS (**A**), 16NS (**B**), and 17NS (**C**) under freezing stress.

**Figure 6 ijms-25-12491-f006:**
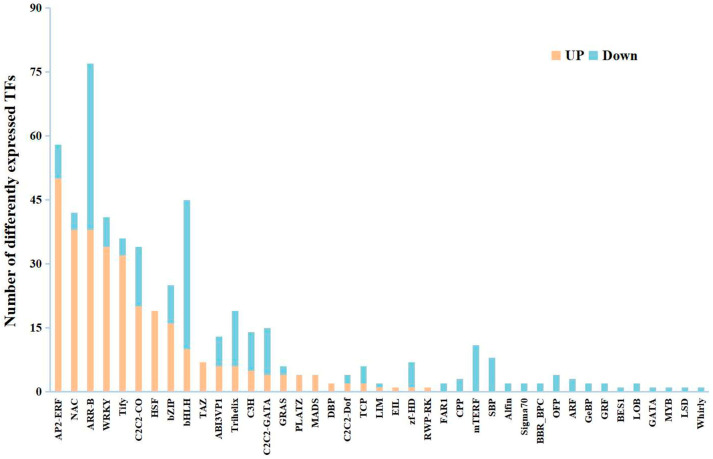
A comprehensive classification of the commonly differentially expressed *TFs* in 15NS, 16NS, and 17NS under freezing stress.

**Figure 7 ijms-25-12491-f007:**
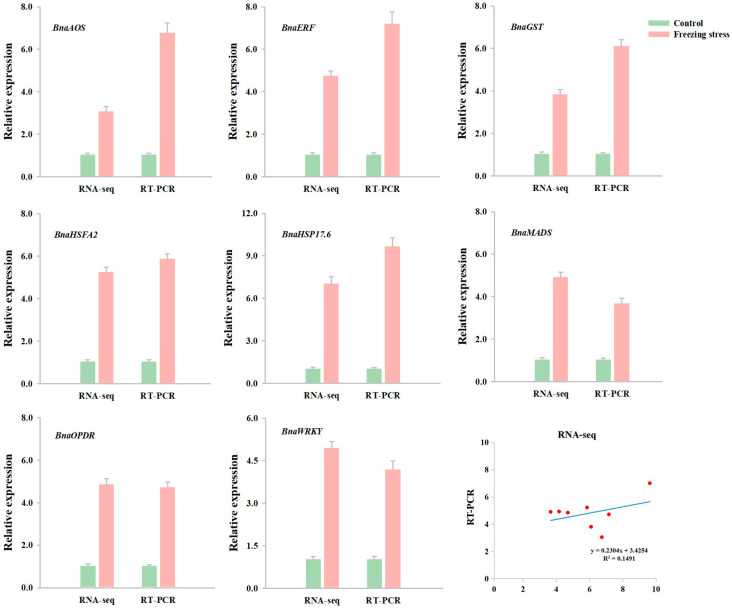
A comprehensive comparison of mRNA expression profiling and RNA-seq quantification in winter rapeseed NTS57 after freezing treatment. The values denote the mean ± standard deviation obtained from three biological replicates.

**Figure 8 ijms-25-12491-f008:**
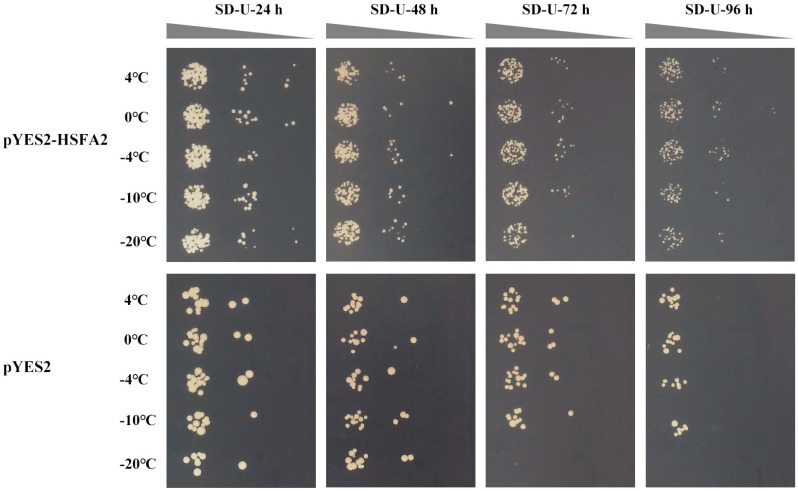
Phenotypic characterization of the *BnaHSFA2* gene in yeast under low temperatures (4, 0, −4, −10, and −20 °C) for 24, 48, 72, and 96 h, respectively.

**Figure 9 ijms-25-12491-f009:**
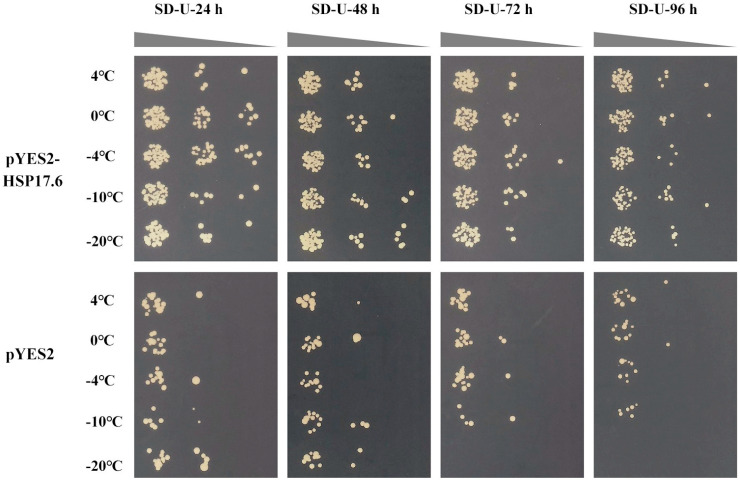
Phenotypic characterization of the *BnaHSP17.6* gene in yeast under low temperature (4, 0, −4, −10, and −20 °C) for 24, 48, 72, and 96 h, respectively.

**Table 1 ijms-25-12491-t001:** Correlation analysis between OWR and physiological indexes.

	POD (U/gmin)	SOD (U/gmin)	CAT (U/gmin)	Proline (mg/g)	Soluble Sugar (ug/g)	Soluble Protein (ug/g)	MDA (μmol/g)	REL (%)
OWR (%)	0.697 *	0.931 **	0.774 *	0.789 *	0.466	−0.921 *	−0.797 *	−0.761 *

**, *p* < 0.01; *, *p* < 0.05.

**Table 2 ijms-25-12491-t002:** Analysis of OWR and physiological indexes.

Year	OWR (%)	POD (U/gmin)	SOD (U/gmin)	CAT (U/gmin)	Proline (mg/g)	Soluble Sugar (ug/g)	Soluble Protein (ug/g)	MDA (μmol/g)	REL (%)
2015	72.3 ± 1.7	25.3 ± 1.4	101.9 ± 6.3	3.06 ± 0.25	0.08 ± 0.01	0.34 ± 0.01	7.48 ± 0.55	6.91 ± 0.38	0.97 ± 0.07
2016	85.3 ± 2.2	30.9 ± 1.7	285.3 ± 9.7	3.97 ± 0.19	0.09 ± 0.01	0.37 ± 0.01	5.84 ± 0.32	5.93 ± 0.19	0.82 ± 0.06
2017	89.5 ± 2.3	33.8 ± 1.5	373.1 ± 16.8	3.92 ± 0.28	0.15 ± 0.01	0.42 ± 0.01	5.72 ± 0.38	3.70 ± 0.18	0.23 ± 0.04

The values represent the mean ± standard deviation derived from three biological replicates.

## Data Availability

The data sets supporting the conclusions of this article are included within the article and its additional files. The sequenced transcriptome raw data have been deposited to the SRA at NCBI with the accession number PRJNA685002.

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
