# Peer review of "Genome-Wide Identification of Freezing-Responsive Genes in a Rapeseed Line NTS57 Tolerant to Low-Temperature"

_ijms, 2024, doi:10.3390/ijms252312491_

Round 1

Reviewer 1 Report

Comments and Suggestions for Authors

Reviewer - Comments

I got acquainted with your manuscript entitled “Genome-wide identification of freezing-responsive genes in a rapeseed line NTS57 tolerant to low-temperature” with great interest. The manuscript presents RNA-seq and correlation analysis between the overwintering rate and the physiological changes of the winter rapeseed freezing-resistant cultivar NTS57. Although the manuscript is well written and explains the DEGs expression analyses in detail. But it couldn’t be accepted in its current form, changes still need to be made to improve the quality of manuscript. I have some comments that need to be addressed before proceeding to the next steps of publication.

1-     What is the motivation of this study, your problem statement is not very compelling please revise and improve it.

2-     Explicitly explain the motivation of your study, don’t just write the generalized statements in the abstract and instruction parts.

3-     Why is the NTS57(NS) cultivar specifically chosen for this study? Is there any research that justifies it selection as a freezing-resistant cultivar?

4-     What are the practical implications of identifying DEGs? How might they be utilized in future breeding programs or genetic modifications to improve freezing tolerance?

5-     There are some typos e.g. double space. Line # 37, 48.

6-     Carefully review the complete manuscript and italicize all the scientific names. Line # 34, 48, 315.

7-     Authors should avoid including generalized statements in the conclusion section. This part should specifically address the findings of your study and offer valuable takeaways for readers. I recommend revising it to ensure clarity and relevance to your research.

Comments on the Quality of English Language

Minor English language improvements are needed.

Author Response

Reviewer comments

I got acquainted with your manuscript entitled “Genome-wide identification of freezing-responsive genes in a rapeseed line NTS57 tolerant to low-temperature” with great interest. The manuscript presents RNA-seq and correlation analysis between the overwintering rate and the physiological changes of the winter rapeseed freezing-resistant cultivar NTS57. Although the manuscript is well written and explains the DEGs expression analyses in detail. But it couldn’t be accepted in its current form, changes still need to be made to improve the quality of manuscript. I have some comments that need to be addressed before proceeding to the next steps of publication.

1- What is the motivation of this study, your problem statement is not very compelling please revise and improve it.

Answer: Thanks for your valuable comments, we have revised the abstract and improved the scientific problems description of this study.

2- Explicitly explain the motivation of your study, don’t just write the generalized statements in the abstract and instruction parts.

Answer: we have revised the abstract and induction (see the lines 7-10 and 48-53).

3- Why is the NTS57(NS) cultivar specifically chosen for this study? Is there any research that justifies it selection as a freezing-resistant cultivar?

Answer: NTS57 (Brassica napus L) is the most cold-resistant winter rapeseed variety in the world. It has strong cold resistance and can survive in the field at −26°C in northwest China. Therefore, the NTS57 was chosen for cold resistance research; Our previous research has shown that it is a freezing-resistant cultivar, and we have added the literature in the manuscript (see the lines 47-49).

4- What are the practical implications of identifying DEGs? How might they be utilized in future breeding programs or genetic modifications to improve freezing tolerance?

Answer: In this study, the DEGs was identified to elucidate the genomic genetic mechanism of strong freezing-resistant winter rapeseed NTS57 so as to provide parental materials for breeding other rapeseed varieties with high quality (low glucosinolates and erucic acid content, high yield) and strong freezing resistance.

5- There are some typos e.g. double space. Line # 37, 48.

Answer: we have deleted double space.

6- Carefully review the complete manuscript and italicize all the scientific names. Line # 34, 48, 315.

Answer: we have italicized all the scientific names (see the lines 19, 20, 35, 49-51, 62, etc.,).

7- Authors should avoid including generalized statements in the conclusion section. This part should specifically address the findings of your study and offer valuable takeaways for readers. I recommend revising it to ensure clarity and relevance to your research.

Answer: Thanks for your valuable comments, we have revised the conclusion (see the lines 400-406).

Reviewer 2 Report

Comments and Suggestions for Authors

The authors conducted freezing-responsive experiments on freezing-resistant rapeseed line NTS57 over different years, explaining the physiological and molecular mechanisms of rapeseed resistance to freezing stress at both physiological and transcriptional expression levels, and identified freezing tolerance-related genes. The research findings provide valuable insights for studying the freezing resistance mechanisms in winter rapeseed. However, there are still some issues in experimental design, data analysis, particularly in the transcriptional data mining. I have raised some questions and provided suggestions for revisions for the authors' consideration.

1. The abstract does not adequately summarize the results; it simply provides a brief overview of the differential gene enrichment analysis results. It is recommended that the authors improve the abstract by incorporating more detailed findings from the study.

2. Line37, “... in a  huge decline...”. There is an extra space between a and huge.

3. Line47, It is advisable to cite references here to demonstrate the development and existence of freezing-tolerant varieties (NTS57).

4. Line75, "p" should be italicized.

5. The section of results 2.1, for the same genotype, the overwintering rate in 2016 and 2017 was significantly higher than that in 2015. Why?

6. The explanation of the experimental design was not clear, making it difficult for me to understand the related results.

7. "it showed a significant positive correlation with overwintering rate". A correlation analysis is needed to establish whether there was a significant positive correlation between these parameters and overwintering rate.

8. Line77-87. There are issues with the terminology used in describing the results; the figure does not indicate significance, yet the text still employs terms related to significance. It is advisable to either annotate the figure with significance markings or present the analysis results in a separate list.

9. It is recommended to combine sections 2.2 and 2.3. The discussion on data quality does not require such extensive elaboration; I believe it can be summarized in just 2-3 sentences.

10. Figure 5. It is suggested to present the differences in comparison and treatment for each year separately, rather than combining them into one.

11. In the section 2.4. The authors should provide a more detailed discussion of the key structural genes within the critical KEGG pathways.

Author Response

Reviewer' comments

The authors conducted freezing-responsive experiments on freezing-resistant rapeseed line NTS57 over different years, explaining the physiological and molecular mechanisms of rapeseed resistance to freezing stress at both physiological and transcriptional expression levels, and identified freezing tolerance-related genes. The research findings provide valuable insights for studying the freezing resistance mechanisms in winter rapeseed. However, there are still some issues in experimental design, data analysis, particularly in the transcriptional data mining. I have raised some questions and provided suggestions for revisions for the authors' consideration.

1. The abstract does not adequately summarize the results; it simply provides a brief overview of the differential gene enrichment analysis results. It is recommended that the authors improve the abstract by incorporating more detailed findings from the study.

Answer: Thanks for your valuable comments, we have revised the abstract. 

2. Line37, “... in a huge decline...”. There is an extra space between a and huge.

Answer: we have deleted the extra space.

3. Line47, It is advisable to cite references here to demonstrate the development and existence of freezing-tolerant varieties (NTS57).

Answer: we have revised the paragraph, and added the reference (see the lines 47-49).

4. Line75, "p" should be italicized.

Answer: we have revised it.

5. The section of results 2.1, for the same genotype, the overwintering rate in 2016 and 2017 was significantly higher than that in 2015. Why?

Answer: There may be an epigenetic phenomenon. It is well-known that the epigenetics is a "memory" behavior. Plants often exhibit enhanced or hyper-sensitivity responses to repeated stresses, resulting in enhanced resistance and adaptation to such stresses (Kinoshita et al. 2014; Martinez-Medina et al. 2016). Kinoshita, T. et al. (2014) Epigenetic memory for stress response and adaptation in plants. Plant Cell Physiol. 55, 1859-1863; Martinez-Medina, A. et al. (2016) Recognizing plant defense priming. Trends Plant Sci. 21, 818-822.

6. The explanation of the experimental design was not clear, making it difficult for me to understand the related results.

Answer: we have revised the full paper in detail to better explain the design of the study.

7. "it showed a significant positive correlation with overwintering rate". A correlation analysis is needed to establish whether there was a significant positive correlation between these parameters and overwintering rate.

Answer: we have added a correlation analysis between overwintering rate and physiological indexes (see the Table 1).

8. Line77-87. There are issues with the terminology used in describing the results; the figure does not indicate significance, yet the text still employs terms related to significance. It is advisable to either annotate the figure with significance markings or present the analysis results in a separate list.

Answer: we are very sorry for our unprofessional description, we have revised the paragraph and annotated the significance markings in Figure 2.

9. It is recommended to combine sections 2.2 and 2.3. The discussion on data quality does not require such extensive elaboration; I believe it can be summarized in just 2-3 sentences.

Answer: we have combined the sections 2.2 and 2.3.

10. Figure 5. It is suggested to present the differences in comparison and treatment for each year separately, rather than combining them into one.

Answer: we have revised this section according to the reviewers’ comments (see the lines 132-149).

11. In the section 2.4. The authors should provide a more detailed discussion of the key structural genes within the critical KEGG pathways.

Answer: we have revised the discussion of the section 2.4 (see the Discussion 3.2). 

Reviewer 3 Report

Comments and Suggestions for Authors

The manuscript entitled “Genome-wide identification of freezing-responsive genes in a rapeseed line NTS57 tolerant to low-temperature” aimed to obtain critical insights into the mechanism of cold tolerance in winter rapeseed under freezing stress. The manuscript is well written, and the research has scientific interest. However, I would like to suggest some minor corrections for the betterment of the article.

  1. The abstract needs to be self-explanatory, and all the abbreviations or shorts need to be introduced (line 11, line 13, etc.). The statement in lines 19 to 21 needs to be rewritten, focusing on the freezing or cold tolerance.
  2. The scientific names and gene names need to be italicized throughout the manuscript.
  3. The abbreviated terms need to be elaborated on at the first mention.
  4. It would be better to enhance the resolution and increase the size of Figure 2 for a clear presentation. We also need to make the legend self-explanatory and elaborate on abbreviated terms.
  5. Please explain or state the uses of the symbol or abbreviated term (15NS, 16NS, etc.) at the first mentioned in the main text (line 114).
  6. If possible, please enhance the resolution of Figure 4.
  7. Proper citation style in text needs to be followed (lines 198 to 200).
  8. Please clearly explain or present your findings in the discussion section, using either a Table, Figure number, or the data (lines 203, 204, etc.).
  9. Please check the superscript letter for lines 375.
  10. It would be better to rewrite the conclusion section for a clear presentation of your concluding remarks.

Overall, there are a few typos in the manuscript that need correction, and proper citation style need to follow.

Author Response

Reviewer's comments

The manuscript entitled “Genome-wide identification of freezing-responsive genes in a rapeseed line NTS57 tolerant to low-temperature” aimed to obtain critical insights into the mechanism of cold tolerance in winter rapeseed under freezing stress. The manuscript is well written, and the research has scientific interest. However, I would like to suggest some minor corrections for the betterment of the article.

1. The abstract needs to be self-explanatory, and all the abbreviations or shorts need to be introduced (line 11, line 13, etc.). The statement in lines 19 to 21 needs to be rewritten, focusing on the freezing or cold tolerance.

Answer: Thanks for your valuable comments, we have revised the abstract and all the abbreviations in the manuscript (see the lines 7-22, 41, 43, 45, 69, 78, 84-85, etc.,).

2. The scientific names and gene names need to be italicized throughout the manuscript.

Answer: we have italicized all scientific names and gene names in the manuscript. 

3. The abbreviated terms need to be elaborated on at the first mention.

Answer: Thanks for your kind reminder, we have revised all the abbreviations (see the lines 11-12, 19-20, 69, 78, 84-85, etc.).

4. It would be better to enhance the resolution and increase the size of Figure 2 for a clear presentation. We also need to make the legend self-explanatory and elaborate on abbreviated terms.

Answer: we have enhanced the resolution of Figure 2.

5. Please explain or state the uses of the symbol or abbreviated term (15NS, 16NS, etc.) at the first mentioned in the main text (line 114).

Answer: we have stated all the abbreviations at the first mentioned in the main text (see the lines 10, 13, and 68).

6. If possible, please enhance the resolution of Figure 4.

Answer: we have enhanced the resolution of Figure 4.

7. Proper citation style in text needs to be followed (lines 198 to 200).

Answer: we have revised it (see the lines 204-205).

8. Please clearly explain or present your findings in the discussion section, using either a Table, Figure number, or the data (lines 203, 204, etc.).

Answer: Thanks for your valuable suggestions, we have revised the paragraph, and added a table for our findings (see the Table 2).

9. Please check the superscript letter for lines 375.

Answer: we have revised it (see the line 390).

10. It would be better to rewrite the conclusion section for a clear presentation of your concluding remarks.

Answer: Thanks for your valuable comments, we have rewritten the conclusion.

Round 2

Reviewer 2 Report

Comments and Suggestions for Authors

I have no other questions.